# Sampling is decisive to determination of *Leishmania (Viannia)* species

**Maxy B. De los Santos**[1]*, **Steev Loyola**[2¤a], **Erika S. Perez-Velez**[3], **Rocio del Pilar Santos**[4], **Ivonne Melissa Ramírez**[5¤b], **Hugo O. Valdivia**[1]

**1** Department of Parasitology, U.S. Naval Medical Research Unit SOUTH (NAMRU SOUTH), Bellavista, Lima, Peru, **2** Facultad de Medicina, Universidad Peruana Cayetano Heredia, Lima, Peru, **3** Departamento Académico de Medicina Humana, Universidad Andina del Cusco, Cusco, Peru, **4** Vysnova, Lima, Peru, **5** Facultad de Ciencias Biológicas, Universidad Nacional Mayor de San Marcos, Cercado de Lima, Lima, Peru

¤a Current address: Grupo de Investigación UNIMOL, Facultad de Medicina, Universidad de Cartagena, Cartagena de Indias, Colombia
¤b Current address: Faculty of Medicine and Health Sciences, University of Antwerp, Antwerp, Belgium
* maxy.b.delossantos2.ln@health.mil

## Abstract

### Background

Accuracy of molecular tools for the identification of parasites that cause human cutaneous leishmaniasis (CL) could largely depend on the sampling method. Non-invasive or less-invasive sampling methods such as filter paper imprints and cotton swabs are preferred over punch biopsies and lancet scrapings for detection methods of *Leishmania* based on polymerase chain reaction (PCR) because they are painless, simple, and inexpensive, and of benefit to military and civilian patients to ensure timely treatment. However, different types of samples can generate false negatives and there is a clear need to demonstrate which sample is more proper for molecular assays.

### Methodology

Here, we compared the sensitivity of molecular identification of different *Leishmania (Viannia)* species from Peru, using three types of sampling: punch biopsy, filter paper imprint and lancet scraping. Different composite reference standards and latent class models allowed to evaluate the accuracy of the molecular tools. Additionally, a quantitative PCR assessed variations in the results and parasite load in each type of sample.

### Principal findings

Different composite reference standards and latent class models determined higher sensitivity when lancet scrapings were used for sampling in the identification and determination of *Leishmania (Viannia)* species through PCR-based assays. This was consistent for genus identification through kinetoplastid DNA-PCR and for the determination of species using FRET probes-based Nested Real-Time PCR. Lack of species identification in some samples correlated with the low intensity of the PCR electrophoretic band, which reflects the low parasite load in samples.

**Data Availability Statement:** All data are within the manuscript and its supporting information. All data is anonymized by codes. The raw data is in S1

Table that permit to replicate the results of our study.

**Funding:** This work was supported by an award from the Armed Forces Health Surveillance Division, Global Emerging Infection Surveillance Branch (AFHSD/GEIS), ProMIS ID 17_N6_1.1.3, 2017-2018, to HOV. The funders had no role in data collection and analysis of study design, decision to publish, or preparation of the manuscript.

**Competing interests:** The authors have declared that no competing interests exist.

## Conclusions

The type of clinical sample can directly influence the detection and identification of *Leishmania (Viannia)* species. Here, we demonstrated that lancet scraping samples consistently allowed the identification of more leishmaniasis cases compared to filter paper imprints or biopsies. This procedure is inexpensive, painless, and easy to implement at the point of care and avoids the need for anesthesia, surgery, and hospitalization and therefore could be used in resource limited settings for both military and civilian populations.

## Author summary

Human cutaneous leishmaniasis affects low-income populations living in places far from health services. The early sampling and detection of the parasite are necessary for timely treatment, however there are no uniform sampling criteria, thus the sensitivity of molecular tests may vary due to various factors such as the type of sample, the time of the disease and the parasite load in the lesion.

In this study, we compared the performance of three sampling methods for molecular identification of the genus and *Leishmania (Viannia)* species in Peru. Several analytical methods, including composite reference standards and latent class models, suggested that lancet scraping might be the best approach for parasite genus detection by kDNA-PCR and for parasite species determination by FRET probes-based Nested Real-Time PCR.

## Introduction

The identification of cutaneous leishmaniasis (CL) is based on clinical presentation and direct parasite identification through microscopy or culture, which could be considered the gold standard [1–3]. However, these methods are less sensitive in comparison with DNA-based molecular detection (approximately 60% and 54% of sensitivity for microscopy and culture, respectively) [4–7].

The World Health Organization does not recommend a specific identification method for leishmaniasis nor a type of sample, but emphasizes the importance of using various parasitological assays, including microscopy, culture, and molecular techniques, to identify the parasite for timely treatment [8]. In contrast, the Infectious Diseases Society of America (IDSA) and the American Society of Tropical Medicine and Hygiene (ASTMH) recommend using full-thickness skin biopsy specimens for histopathology and cultures and multiple diagnostic approaches to maximize the likelihood of a positive result [9].

Multi Locus Enzyme Electrophoresis (MLEE) has been considered the gold standard for taxonomy and identification of *Leishmania* species [10–12]. However, this technique requires the isolation and mass culture of parasites to obtain sufficient quantities of proteins for enzymatic assays, which is very time-consuming, labor–intensive, expensive and technically demanding [13]. As an alternative method to MLEE, many PCR-based protocols have been developed, these assays only need to collect tissue using different methods and instruments consider invasives such as punch biopsies, syringe aspirate, scalpel, or lancet scraping, and other non-invasives like filter paper impression and rotating a swab, but with variable sensitivities. Molecular assays with higher sensitivity allows not only to characterize the parasite at the genus or species level, but also to quantify the parasite load in the lesion, evaluate the course of

experimental drug therapies, identify virulence factors or drug resistance markers, and support in surveillance and control programs for leishmaniasis [14].

While different PCR-based methods including PCR-RFLP, nested-PCR, real time PCR, PCR-ELISA, oligo chromatography-PCR have been developed to improve the sensitivity of the detection of parasite DNA, few studies have been conducted to assess the effect of the sampling method and site in diagnosis [15–17]. These studies have demonstrated better results when samples are taken from the base and center of the ulcer, and suggested the use of less invasive methods instead of biopsy to avoid local anesthesia, risk of bleeding and infection [18,19]. Furthermore, it has been suggested that the parasite load varies according to the sampling method and site, hence affecting the ability to detect *Leishmania* [20].

It has been suggested that a single smear sample rather than more invasive samples is sufficient to obtain a positive PCR result [21]. Previous reports compared the results obtained by PCR with less sensitive methods, such as microscopy, culture and rapid tests [22–25] as well as the results obtained using different types of PCR [5,26,27]. However, few studies have evaluated the performance of PCR-based molecular tests with different types of clinical samples, invasive and non-invasive, including different physical sample supports such as filter paper and Giemsa-stained slides, with wide variability in sensitivities or percentage of positives, without a clear consensus on the easiest, fastest and most painless collection method for the benefit of military and civilian patients that guarantees accurate identification of the parasites and timely treatment [7,16,28,29].

In the present study, the performance of two molecular assays, kDNA-PCR for genus identification [30] and FRET probes-based Nested Real-Time PCR for species determination [31], were compared using a group of three types of samples, biopsy by punch (Bx), imprint by filter paper (FP) and scraping by lancet (L), each taken from the same patients. Differences in the sensitivity of the identification and determination of *Leishmania (Viannia)* species across the sampling types was evaluated using different models of composite reference standards (CRS) and latent class models (LCM). Likewise, we observed that less intensity of the amplification band in kDNA-PCR were associated with the inability to determine species of the parasite, thus we evaluated the variations in the parasite load that could explain this phenomenon. The molecular results have also been compared with parasite isolation by culture, which was achieved in a subset of patients.

## Methods

### Ethics statement

This study was approved by the Institutional Review Board of the U.S Naval Medical Research Unit SOUTH (NAMRU SOUTH) in compliance with all applicable federal regulations governing the protection of human subjects (NMRCD.2007.0018).

Trained personnel obtained written informed consent from each enrolled patient and completed a survey to collect demographic, clinical and epidemiological data.

### Study population

Clinical samples for this study were selected from a set previously collected between 2013 to 2016 from individuals older than five years-old, non-pregnant in the case of women, who presented with suspected cutaneous leishmaniasis lesions without previous history of treatment in the last six months. All clinical samples included in this study were obtained from the Hospital Militar Central in Lima, and health facilities in Iquitos city (Laboratorio Referencial de Salud Pública, Hospital Iquitos César Garayar García) and Madre de Dios region (Delta 1 Bajo Puquiri, Jorge Chávez Health Center, Nuevo Milenio Health Center, and Laboratorio

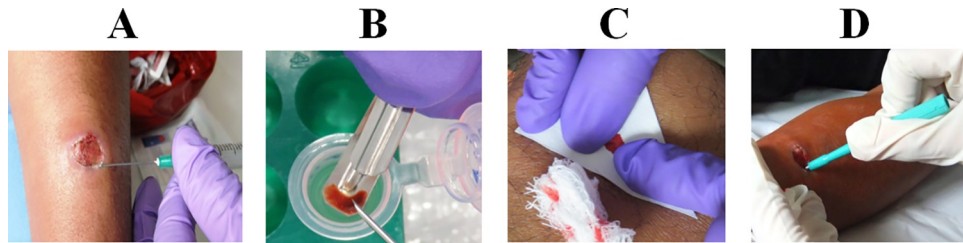

**Fig 1. Sampling methods used per patient.** A. Aspirate for culture, B. Scraping by lancet; C. Imprint by Whatman filter paper; D. Biopsy using a 3 mm sterile punch. The photographs were taken by MBD with a Fujifilm FinePix JX250 14 MP Digital Camera.

Referencial de Salud Pública de Madre de Dios). Additionally, blood samples were collected from healthy individuals from non-endemic area as negative control. The different samples obtained were coded and processed blindly, without knowledge of previous or by-pair results from their arrival at the laboratory, to avoid analytical biases during all procedures and stages of this study.

## Parasites culture

After the lesion was disinfected with 70% ethanol and scabs were removed with a gauze, a 21G, 3cc, 1 1/2" syringe with needle containing 0.1 ml of sterile saline solution (1% gentamicin 10 mg/ml, 1% 5-Fluorocytosin 5mg/ml) was inserted to the internal border or the center of the ulcer. Then the syringe was slightly rotated (Fig 1A), and the fluid was aspirated and ejected into a culture tube with USAMRU sterile blood agar. Parasite growth was monitored for two weeks until promastigotes were visible at 400X in an inverted microscope. Cells were cryopreserved in liquid nitrogen for further studies.

## Lesion sampling

This study selected subjects that had three different types of clinical samples for molecular assays. Subjects with only one or two types of samples were excluded. After aspirating the lesion for parasite isolation, a clinical sample from the internal border of the lesion was collected by scrapping using a sterile lancet (L) which was later stored in a 1.5 ml microtube with absolute ethanol (Fig 1B). Then, a piece of filter paper (FP, Whatman 3 Qualitative) was gently pressed over the lesion to allow the adherence of fluids and lesion material (Fig 1C). The filter paper was dried at room temperature, cut, and stored in a 1.5 ml microtube with absolute ethanol. Finally, local anesthesia with xylocaine at 5% was injected in the internal border of the lesion to take a tissue biopsy (Bx) with a 3 mm punch biopsy instrument (Fig 1D), which was stored in a 1.5 ml microtube with absolute ethanol. All samples were sent at room temperature to NAMRU SOUTH facilities in Lima and they were stored at -20˚C until DNA extraction.

## DNA extraction

Nucleic acids from the three types of clinical samples (L, FP and Bx) were extracted using the DNeasy Blood & Tissue kit (QIAGEN, Maryland, USA) following the manufacturer's instructions with modifications according to the sample type [32]. Briefly, to obtain optimum DNA yield and quality, we used up to 25 mg of tissue from Bx and L and 1 cm of diameter for FP cut in small pieces. Biological material from L was separated with a 21G sterile needle and minced with the same needle in a microfuge tube. Then, the tube was centrifuged for 5 minutes at 14,000 rpm, supernatant was discarded, and the pellet was dried at 56˚C. The material from FP

was extracted by drying the paper on a clean microscopy slide, cut in tiny pieces with a clean blade and heat in 180 μL of ATL buffer at 85°C for 10 minutes in a clean 1.5 ml microcentrifuge tube. Finally, tissue obtained by punch biopsy was washed with 0.5 ml of 1X sterile phosphate buffer saline (PBS 1X) to eliminate traces of ethanol, then the tissue was put on a clean microscopy slide and minced with a clean blade. The material obtained was put in a new microcentrifuge tube. All sample types were digested overnight at 56°C with 20 μl of Proteinase K in 180 μl of ATL buffer. Then, all samples were heated at 70°C for 10 minutes with AL lysis buffer and purified at room temperature by centrifugation at 8,100 rpm through silica membrane DNeasy spin columns. Then, the columns were washed with buffers AW1 an AW2 based in guanidine hydrochloride. Extracted DNA from all the three sample types was eluted by centrifugation (8,100 rpm) at room temperature for 2 minutes with AE buffer (10 mM Tris-Cl, 0.5 mM EDTA; pH 9.0) and then stored at -20°C until use.

### Identification and characterization of *Leishmania* species

*Leishmania (Viannia)* genus was identified using PCR targeting the minicircle kinetoplast DNA (kDNA-PCR) with specific primers MP1L and MP3H following conditions previously described [30]. *Leishmania* species were detected by FRET probes-based Nested Real-Time PCR based in the identification of single nucleotide polymorphisms (SNP) of the Mannose Phosphate Isomerase (MPI) and 6-Phosphogluconate Dehydrogenase (6PGD) genes using the LightCycler 480 II system (Roche Diagnostics, Basel, Switzerland). LC 480 Genotyping Master and protocol for melting curve analysis with HybProbe format from LC 480 Software v1.5 were used as previously described [31].

### Evaluation of kDNA-PCR amplification band intensity

Variation in 70 bp kDNA-PCR amplification band intensity was assessed by 2.5% agarose gel electrophoresis with GelRed Nucleic Acid gel stain in 1X Tris–Acetate-EDTA buffer (TAE buffer). Invitrogen 100 bp DNA Ladder was used for identification of size amplification band. The amplification bands from different sampling methods were compared with the intensity of the positive control band using *Leishmania (Viannia) braziliensis* LTB300 strain in a Chemi-Doc MP Imaging System with Image Lab software v5.1 (BIO-RAD).

### Quantitative PCR

Parasite load was determined on the three paired types of samples for each patient (Bx, FP, L) by quantitative PCR (qPCR) using SYBR Green I Master (Roche). The standard curve to quantify the number of parasites was prepared with 8 dilutions of genomic DNA of *Leishmania (Viannia) braziliensis* MHOM/BR/84/LTB300 strain covering a range of $5 \times 10^{-3}$ to $5 \times 10^{4}$ parasites based on an equivalence of 83.15 fg to one parasite [33]. A standard curve of endogenous retrovirus 3, ERV3 for human cells covering a range of 160 to $20 \times 10^{3}$ copies was used to normalize [34].

Quantitative PCR was performed in 25 μl with 0.2 μg/μl DNA sample, 200 μM of each primer (MP3H and MP1L for kDNA-qPCR or ERV3-F and ERV3-R for ERV3-qPCR) in a LightCycler 480 II (Roche) as previously described [17,33].

Parasite numbers were normalized in relation to the human endogenous target based on the equation: $L^{*}10^{6}H$ = *Leishmania* parasites per $10^{6}$ human cells = [number of parasites/(copy number of ERV3/2)]$^{*}$ $10^{6}$ as previously described [33].

All procedures described so far are summarized in a flowchart for further understanding (S1 Fig).

## Tests accuracy estimation

Two approaches were used to mitigate misclassification induced by the use of imperfect tests: composite reference standards (CRS) and latent class models (LCM) [35].

Two CRS were constructed for classifying individuals using the "OR" rule [36]; CRS1, combined results of the kDNA-PCR in the three types of samples and results of the cell culture, and CRS2 combined results of CRS1 and FRET probes-based Nested Real-Time PCR using the three types of samples. Specifically, individuals were classified as *cases* if any of the previously listed assays yielded a positive result. Besides, *non-cases* were individuals who had negative results in all assays.

In LCM, conditional independence was assumed because molecular tests are nucleic acid amplification techniques and the cell culture is based on the parasite's growth and multiplication in a culture medium [37,38]. Based on the observed results of each molecular detection and cell culture, a latent class analysis (LCA) was used to compute the probability of each individual being classified as a *case* or *non-case* [35,39–41]. In the LCA, each test was used as an indicator variable. A total of six models of LCA (LCM) were constructed; three using results from all kDNA-PCR and the cell culture (LCM1-LCM3), and other three using results from all molecular assays (kDNA-PCR and FRET probes-based Nested Real-Time PCR) and cell culture (LCM4-LCM6). As previously described [41–44], basic two-class models were built to classify individuals into two exclusive and exhaustive groups (LCM1, LCM2, LCM4, and LCM5). Parasite load was included as a covariate in LCM2 and LCM5 because it can influence the results of the tests. The use of parasite load (herein: intensity of PCR bands in agarose gel) as a covariate in the model was based on a heuristic approach that described the relationship between high parasite load and a high probability of positive test results [45]. Finally, three-class models (LCM3 and LCM6) were built as exploratory models, as described elsewhere [42,46], to assess the improvement in model fit.

## Statistical analysis

Absolute and relative frequencies were used to describe categorical variables. Median (p50) and interquartile range (IQR) were used to summarize numerical variables. Absolute values of parasite load obtained from the qPCR assays were converted to logarithms [function $log$(parasite load)] because the load displayed a non-normal distribution. Then, the parasite load for each paired sample type was plotted individually, and the overall parasite load by sample type and gel electrophoretic band intensity was plotted using box-and-whisker plots. To assess differences in parasite load between paired samples, the Friedman and Dunn's multiple comparison test was used.

LCM1 and LCM2 were implemented using logistic models, while the other ones used the *probit* function due to non-convergence with logistic functions. The rationale for choosing the LCA-derived reference standard was based on parsimony and fit. Specifically, the best LCM was selected using a parsimony criterion (fewer degrees of freedom) and various statistical criteria based on log-likelihood (LL), the Akaike's information criterion (AIC), and the adjusted Bayesian information criterion (aBIC) [41,46]. The response pattern of the best LCM was summarized according to the combination of test results and classes.

The McNemar's chi-square test was used to assess differences in the classification of individuals, and the Cohen's Kappa coefficient was used to assess the agreement between reference standards and assay types in each group of samples. Tests performance was estimated in 2 x 2 contingency tables using the groups and classes defined by the CRS and the best LCM, respectively. The sensitivity and specificity, as well as their respective 95% exact binomial confidence intervals (95%CI), were estimated. A test for receiver operating characteristic (ROC) area

equality with Sidak's adjustment for multiple comparisons was also used. Analyses were performed in Stata v17 (StataCorp. 2021. Stata Statistical Software: Release 17. College Station, TX: StataCorp LLC.) and R/Rstudio v2022.07.2 for Windows, and $p < 0.05$ was considered as significant.

## Results

### Characteristics of the study population

From a group of patients previously enrolled between 2013 to 2016 suspicious cutaneous leishmaniasis cases that had at least three different sampling methods were selected (105 patients, 315 samples of Bx, FP and L). Participants came from the Peruvian departments of Amazonas, Ayacucho, Cajamarca, Cusco, Huánuco, Junín, La Libertad, Lima, Loreto, Madre de Dios, San Martín and Ucayali which are endemic areas where we conduct leishmaniasis surveillance. Only three subjects were women, 75 were military and 30 were civilian, with a median age of 29 years-old (Range: 17–78). Most participants (80%) only had a single lesion (Range: 1–8) with a median size of 1.5 cm in diameter (Range: 0.2–15) and only one participant had mucosal lesion with perforation of the nasal septum (Tables 1 and S1. The median time of disease evolution prior to enrollment was two months ranging from one week to eight years, and 10 participants could be classified as chronic cases (more than six months of disease). It was not observed that any clinical trait (Tables 1 and S1) was associated to the molecular results obtained with the three types of samples ($p > 0.050$).

### Four *Leishmania (Viannia)* species were identified

From the 105 enrolled patients, kDNA-PCR identified *Leishmania (Viannia)* parasites in 85 of them whereas the FRET probes-based Nested Real-Time PCR identified species in 69 of them, most cases were identified as *Leishmania (Viannia) braziliensis* (50 patients, 72.5%) from the departments of Ayacucho, Cusco, Junín, Loreto, Madre de Dios and Ucayali. Additionally, there were three other species identified: *L. (V.) guyanensis* (11 patients, 15.9%) in samples from the departments of Ayacucho, Cusco, Junín, Loreto and San Martín; *L. (V.) lainsoni* in four patients from the departments of Amazonas, Cusco and Loreto (5.9%); one case of *L. (V.) peruviana* from Amazonas (1.4%); and three from Cusco and Loreto showed a putative pattern of *L. (V.) braziliensis/L. (V.) peruviana* hybrids (4.3%) (Fig 2). In 16 patients from Amazonas,

**Table 1. Clinical characteristics of enrolled participants.**

| Variable | n (%) |
|---|---|
| Male | 102 (97.1) |
| Age (years)* | 29 (14.0) |
| 1 lesion | 84 (80.0) |
| 2 lesions or more | 21 (20.0) |
| Time of disease (months)* | 2.0 (3.0) |
| Lesion size (cm)* | 1.5 (1.0) |
| Enrolled in 2013 | 16 (15.2) |
| Enrolled in 2014 | 27 (25.7) |
| Enrolled in 2015 | 37 (35.2) |
| Enrolled in 2016 | 25 (23.8) |

*Median (IQR)

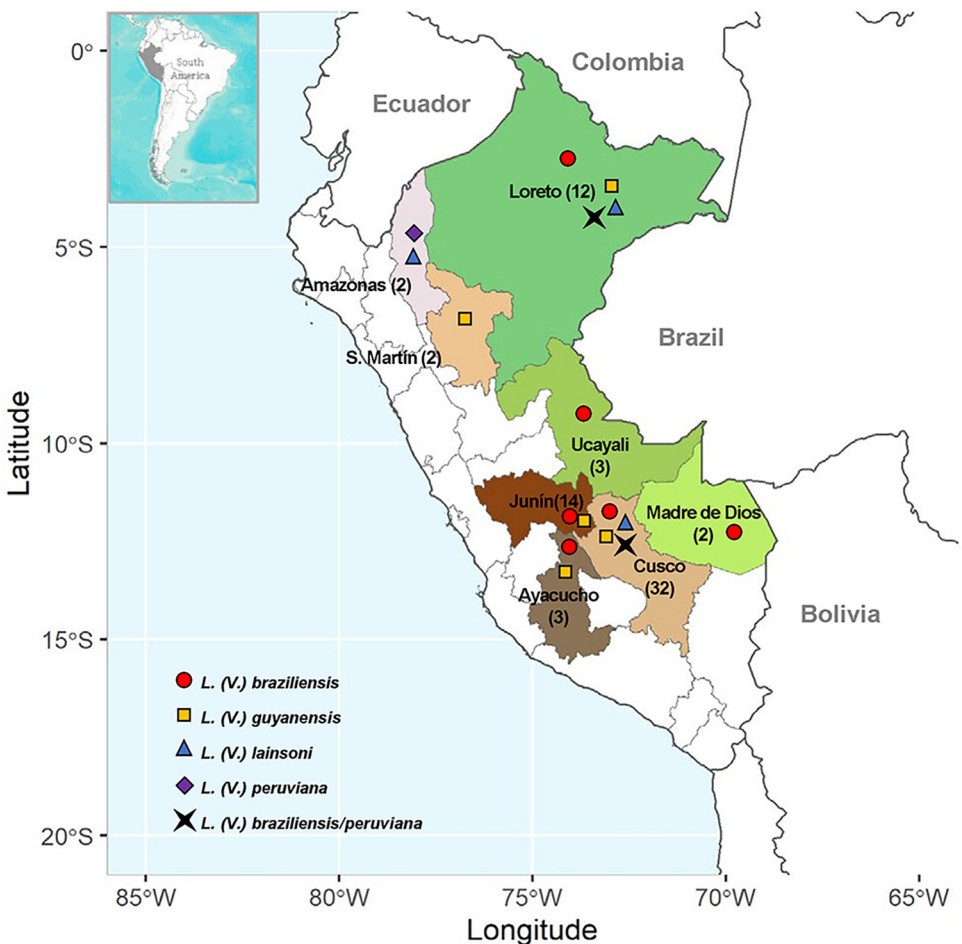

**Fig 2. Distribution of *Leishmania (Viannia)* species isolates from eight departments of Peru.** Numbers between parentheses represent participants per department. Caption: This map was created on R using open data obtained from GADM database of Global Administrative Areas, version 3.6. www.gadm.org and the Sf package (https://cran.r-project.org/package=sf) with GPL-2 license (https://cran.r-project.org/web/licenses/GPL-2) and the ggplot2 package (https://ggplot2.tidyverse.org/index.html) with MIT license (https://ggplot2.tidyverse.org/LICENSE.html).

Cusco, Huánuco, Junín, La Libertad, Lima, Loreto and Ucayali, it was not possible to identify the species of the infecting parasite, which was named *Leishmania* spp.

## Sample type affected the identification of *Leishmania (Viannia)*

The analysis of 315 clinical samples (105 patients) showed that the sampling method had an important influence in the identification of *Leishmania (Viannia)*. In terms of genus detection, the kDNA-PCR assay showed 1.9% and 4.8% more positives when imprint by filter paper (FP) and scraping by lancet (L) were used in comparison to biopsy by punch (Bx), (Table 2). Species identification by FRET probes-based Nested Real-Time PCR using L identified 9.5% more positives than FP and 8.6% more positives than Bx (Table 2).

The kDNA-PCR amplification band presented lower intensity in those samples that did not amplify to determine species. The qPCR showed that the *log (parasite load)* of 159 paired samples from 53 individuals, was significantly higher in the FP (p = 0.031) and L samples (p = 0.033) compared to the Bx samples (Table 2). This was also observed in the parasite load of the FP and L samples compared to the Bx samples.

**Table 2. Positivity and parasite load by assay and sample type, and composite reference standard (N = 105).**

| Type of assay/analysis | n (%) |
|---|---|
| **Cell culture** | 41 (39.1) |
| **kDNA-PCR** | |
| Biopsy | 79 (75.2) |
| Filter paper | 81 (77.1) |
| Lancet | 84 (80.0) |
| **Nested RT-PCR** | |
| Biopsy | 58 (55.2) |
| Filter paper | 57 (54.3) |
| Lancet | 67 (63.8) |
| **qPCR (parasite load)*** | |
| Biopsy | 9.6 (7.3) |
| Filter paper | 10.0 (5.7) |
| Lancet | 9.9 (6.5) |
| **CRS** | |
| CRS1 | 85 (81.0) |
| CRS2 | 85 (81.0) |

*Log(Leishmania parasites per $10^6$ human cells)

### Higher parasite load was associated with positive results in the FRET probes-based Nested Real-Time PCR

Regardless of the type, if a sample amplified by kDNA-PCR produced a single band with a weak or very weak intensity in agarose gel electrophoresis compared to the positive control (Fig 3A), the likelihood of non-amplification in either or both genes (Mannose Phosphate Isomerase and 6-Phosphogluconate Dehydrogenase) in the FRET probes-based Real-Time PCR increased. Thus, in these samples, the identification of *Leishmania (Viannia)* species was not achieved (Fig 3B) and could correlate with a lower parasite load in comparison to the positive control and other samples with a strong positive band. Similarly, the analysis of the

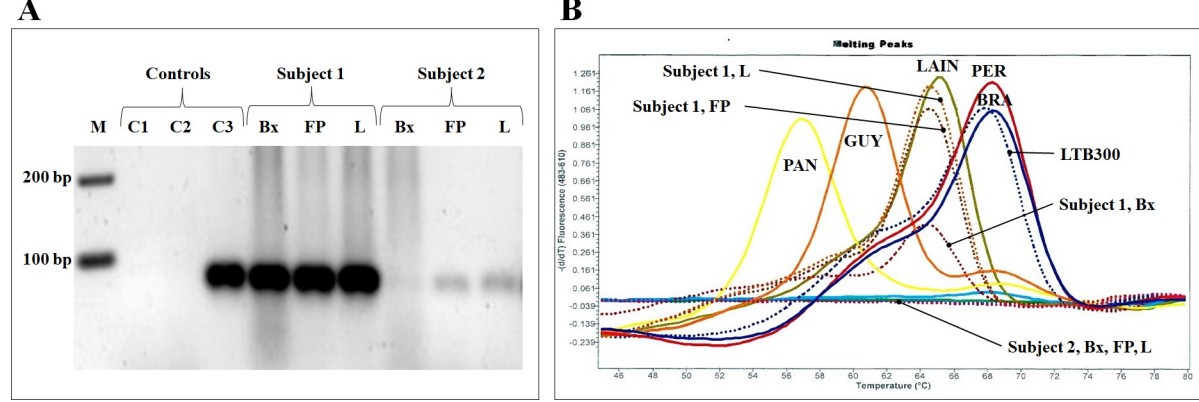

**Fig 3. Weak electrophoretic band intensity in kDNA-PCR correlated with the inability to determine species by FRET probes-based Nested Real-Time PCR.** A) Electrophoresis in 2.5% agarose gel. Legend: C1, distilled water PCR grade; C2, human DNA as negative control; C3, *L. (V.) braziliensis* LTB300 DNA as positive control; Bx biopsy samples; FP filter paper samples; L lancet samples. B) Melting peaks of 6PGD gene obtained for FRET probes-based Nested Real-Time PCR. Legend: PAN, *L. (V.) panamensis*; GUY *L. (V.) guyanensis*; LAIN, *L. (V.) lainsoni*; PER, *L. (V.) peruviana*; BRA, *L. (V.) braziliensis*.

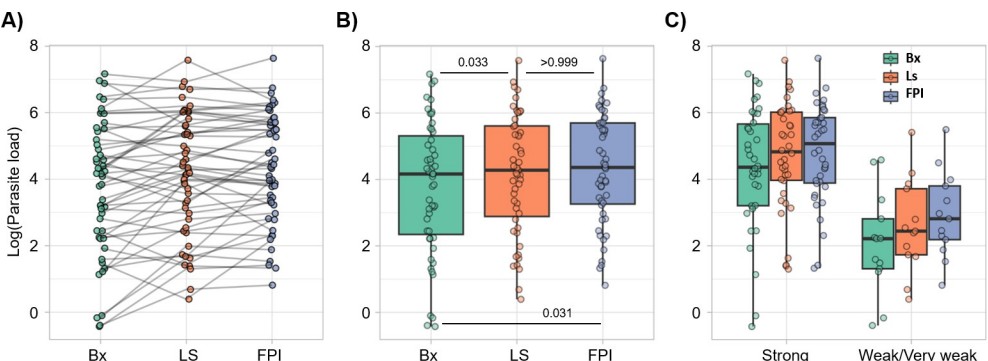

**Fig 4. Parasite load is related with intensity of kDNA-PCR electrophoretic band.** A) Individual parasite load in paired samples. B) Overall parasite load by sample type. C) Comparative parasite load by type of samples and intensity of electrophoretic bands. Bx: Biopsy; FP: Filter paper imprints; L: Lancet scraping.

parasite load obtained after the qPCR showed differences between the clinical samples, being significantly higher in FP (p = 0.031) and L (p = 0.033) in contrast to Bx (Fig 4B). The same pattern was observed between samples with a strong versus weak or very weak band for kDNA-PCR, where the high parasite load corresponded to the strong bands in all types of samples in contrast to the lowest values of parasite load in the weak or very weak (Fig 4C).

## Classification of individuals according to CRS and LCM

According to both CRS, the frequency of cases in the study was 81.0% (85/105 cases, Table 2). Individual classification did not differ based on the CRS used (p = 1.000), resulting in a very high degree of agreement between the two CRS (Kappa = 1.000, 95%CI: 1.000–1.000). Based on these findings, subsequent analyzes included only CRS1 because it contained the fewest tests in its construct (Table 2).

LCM1 was identified as the most parsimonious model due to its lower degrees of freedom compared to the other models, whereas LCM6 was the most complex (Table 3). According to the log-likelihood of the model, the AIC and the aBIC, the LCM1 and LCM2 models were the best and allowed to classify 84 (80%) individuals as *cases* (Table 3, Class 2 column). There was no difference in the classification of the individuals (p = 1.000), and both models exhibited high agreement between them (Kappa = 1.000, 95%CI: 1.000–1.000).

**Table 3. Different models of Latent Class Analysis (N = 105).**

| Latent Class Models | LL(model) | df | AIC | aBIC | Class 1 | Class 2 | Class 3 |
|---|---|---|---|---|---|---|---|
| | | | | | % (n) | | |
| 3 kDNA-PCR + cell culture | | | | | | | |
| (LCM1) 2-class | -151.99 | 9 | 321.98 | 345.87 | 20.0 (21) | 80.0 (84) | - |
| (LCM2) 2-class + load | -146.71 | 10 | 313.42 | 339.95 | 20.0 (21) | 80.0 (84) | - |
| (LCM3) 3-class | -147.53 | 15 | 323.07 | 360.23 | 20.0 (21) | 40.9 (43) | 39.1 (41) |
| 3 kDNA-PCR + 3 RT-PCR + cell culture | | | | | | | |
| (LCM4) 2-class | -622.72 | 15 | 1275.43 | 1315.24 | 32.4 (34) | 67.6 (71) | - |
| (LCM5) 2-class + load | -270.6 | 16 | 573.19 | 615.66 | 41.9 (44) | 58.1 (61) | - |
| (LCM6) 3-class | -292.83 | 23 | 611.66 | 672.7 | 19.1 (20) | 23.8 (25) | 57.1 (60) |

LL(model): log-likelihood of the model. df: degrees of freedom. AIC: Akaike's information criterion. aBIC: Adjusted Bayesian information criterion. LCM: Latent class model. RT-PCR: FRET probes-based Nested Real-Time PCR.

**Table 4. Eight different patterns grouped in two classes were generated with the model LCM2.**

| kDNA-PCR | | | Culture | Class 1 | Class 2 |
|---|---|---|---|---|---|
| Bx | FP | L | | n | n |
| Neg. | Neg. | Neg. | Neg. | 20 | 0 |
| Neg. | Neg. | Pos. | Neg. | 1 | 0 |
| Neg. | Pos. | Pos. | Neg. | 0 | 5 |
| Pos. | Neg. | Pos. | Neg. | 0 | 1 |
| Pos. | Neg. | Pos. | Pos. | 0 | 2 |
| Pos. | Pos. | Neg. | Pos. | 0 | 1 |
| Pos. | Pos. | Pos. | Neg. | 0 | 37 |
| Pos. | Pos. | Pos. | Pos. | 0 | 38 |

Bx: Biopsy; FP: Filter paper imprints; L: Lancet scraping; Neg: Negative results; Pos: Positive results

In contrast to LCM1 (the basic model), LCM2 had a better fit, included parasite load as a covariate, and reproduced the same distribution of individuals by class as LCM1 (Table 3). The LCM2 model generated eight different patterns of results. The first two patterns were grouped in Class 1 with 21 individuals; 20 of which tested negative by kDNA-PCR and cell culture in all samples, while one individual tested positive for the lancet used in the kDNA-PCR (Table 4). The other patterns grouped 84 individuals in Class 2 since they presented at least two positive results in the samples analyzed by kDNA-PCR and one positive or negative result in the cell culture (Table 4).

When comparing CRS1 and LCM2, a nearly identical grouping was observed; 20 individuals classified as *non-cases* by both approaches, 84 individuals as *cases* by both approaches, and one individual was classified as a *non-case* by LCM and as a *case* by CRS. Despite the discrepancy, the standard references were comparable and exhibited a high level of agreement (p = 0.317, Kappa = 0.970, 95%CI: 0.911–1.000). To evaluate the diagnostic performance of each index test, both CRS1 and LCM2 were used in the subsequent analyses.

## CRS and LCM demonstrated no significant differences between samples for our molecular assays

When evaluating the discriminatory capacity of the kDNA-PCR or the FRET probes-based Nested Real-Time PCR through the area under the ROC curve, it did not show significant differences (p>0.050) regardless of the type of sample (Bx, FP, or L) or reference standard evaluated (LCM or CRS). However, in all scenarios where kDNA-PCR were used, there was a significant increase in sensitivity and the area under the ROC curve (p<0.050) when compared to the FRET probes-based Nested Real-Time PCR and cell culture. Although the estimated sensitivity with the LCM2 was higher than the CRS, the difference was not significant (p>0.050). Finally, regardless of the reference standard used (LCM or CRS), the sensitivity of the index tests was higher when lancet scraping samples were used. In general, no significant differences in specificity were found between the evaluated tests (Fig 5).

## Discussion

This study demonstrates that different sampling methods can affect the discrimination capacity of molecular tests for the identification of parasite and determination of *Leishmania (Viannia)* species. Likewise, the ability to detect the parasite by PCR amplification is associated with the parasite load in the samples. These observations were independent from clinical

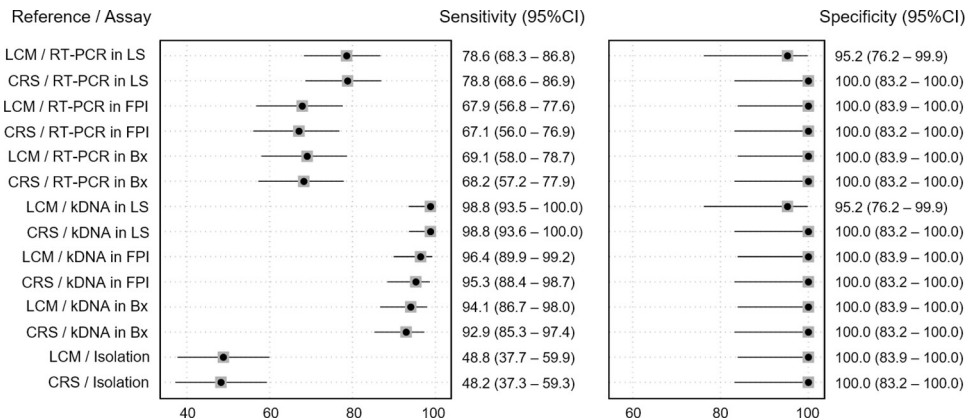

**Fig 5. Sensitivity and specificity for each reference and technique in relation to the type of biological sample.**
LCM: Latent class model; CRS: Composite reference standard; RT-PCR: FRET probes-based Nested Real-Time PCR;
L: Lancet scraping; FP: Filter paper imprints; Bx: Biopsy.

parameters as age of patients, time of disease evolution (considering <6 months for acute cutaneous leishmaniasis and ≥6 months for chronic cutaneous leishmaniasis, like [47]), size and location of lesions, number of lesions, and the *Leishmania* species identified in this study.

Similar to our previous report [31], the molecular assays allowed us to identify the species of *Leishmania (Viannia)* in most clinical samples. Interestingly, despite of not being the objective of the study, our results confirmed the wide geographical distribution of *Leishmania* species observed in previous reports [48,49], possibly due to an increase in human migration and environmental changes observed in the recent years [50]. Similar to previous studies, *L. (V.) braziliensis* was the most prevalent species followed by *L. (V.) guyanensis* and *L. (V.) lainsoni* [48,51,52].

The detection of the genus *Leishmania* by kDNA-PCR, although it does not show significant differences between sample types, does not mean that there were no differences between the sensitivities obtained for each type of sample, with the lancet being similar to filter paper and superior to biopsies. This could be explained by the greater amount of parasite DNA in the superficial layers of the skin taken with the lancet and filter paper (72) unlike a biopsy that can collect a greater amount of skin tissue and therefore human DNA (15). Moreover, the need of local anesthesia and surgery with pain and distress for patients become impractical the biopsy in field conditions [17]. Therefore, we encourage the use of non-invasive and outpatient methods such as lancet scraping or filter paper impressions, with results similar to invasive methods such as biopsy, with the advantage of avoiding distress, pain and in some cases hospitalization in civilian and military patients.

Previous studies using scrapping employed a number 11 sterile scalpel or a number 15 surgical blade [5,7,15], but we prefer the use of a sterile aluminum lancet given the possibility of bleeding due to mishandling of the scalpel. Another advantage of lancet scraping was obtaining sufficient material for our molecular assays rather than invasive samples (aspirates, biopsies), similar to the results of a meta-analysis suggesting that smears are very sensitive tools for the detection of *Leishmania* instead of invasive biopsies or aspirates [21]. Additionally, the smear obtained by lancets can be used in microscopy [5] and put in filter paper for DNA extraction and detection by PCR [53].

Regarding filter paper, we prefer to work with Whatman 3 Qualitative because its coarse porosity matches that of previous studies [4,5,7,53,54]. A simple and soft pressure on the lesion

allowed obtaining enough tissue fluid and recovering DNA from the parasites without the need to put biopsy fragments or smears on the filter paper [53,54].

Noteworthy, for more than 20 years studies have suggested the use of scraping with lancet or scalpel as an alternative to biopsies for the diagnosis of cutaneous leishmaniasis by PCR [16,55] to avoid the need of personnel specialized in surgery and iatrogenic bacterial infections [16,56]. Despite this evidence, the systematic use of biopsies continued, especially for comparison of PCR with microscopy [57,58] or to study the torpid evolution of the disease [59]. Furthermore, biopsies stored in formalin require trained personnel in histology to identify the parasite, but the determination of species is not possible on this type of sample.

Another alternative non-invasive sampling methods like swabs, tape strip disc and cytology brushes may be particularly useful, especially for lesions located on mucosal areas and in body zones with higher sensitivity [18,60,61] or in the case of children [62–64]. Similar to lancets and filter papers, these methods reduce the risk of bleeding and exposure of health personnel to body fluids [4,61,65].

There is a need for a standardized sampling system in function of the clinical characteristics of the patients. For example, low parasite load of *Leishmania (Viannia) braziliensis* is observed in the chronic cutaneous disease (lesions persist for more than six months) [66], but the criteria to establish a division between acute and chronic disease are missing, since more than three months and 12 months were previously was considered chronic cutaneous leishmaniasis [4,7,15,17].

The high sensitivity and apparent homogeneity from kDNA-PCR to detect the parasite, independently from the type of samples used, could be explained by the large number of copies of the minicircle fragment in the kinetoplastid DNA to be amplified and the interspecific constitutive homogeneity of the amplified minicircle sequence [67]. Other targets have been analyzed principally in Old World *Leishmania* species through PCR-RFLP, DNA sequencing and melting analysis, but the sensitivity is highly variable [68].

Melting curve analysis by Real-Time PCR has been used to identify *Leishmania* species in the Old World [54,69,70]. In a previous work, we demonstrated a sensitivity of 92% and a specificity of 77% of FRET probes-based Nested Real-Time PCR which was much higher than identification by culture, microscopy and leishmanin skin test in Peruvian clinical samples [31]. The current study showed no significant differences between types of samples compared, and like kDNA-PCR, when the punctual estimates of the sensitivities are taken, this molecular tool shows higher values when the lancet is used in contrast to filter paper and biopsy.

A limitation in all the tests to identify the parasite, even by molecular methods, is the presence of false negatives. This study demonstrated an association between false negatives for the determination of parasite species by FRET probes-based Nested Real-Time PCR and the reduction in the intensity of the amplification band by kDNA-PCR as described in a previous study that demonstrates that it is not possible to determine species using different targets by PCR when the kDNA-PCR results show a low intensity of the amplified band [20]. Another limitation of this method for species determination is the amount of amplified target DNA, basically only two gene copies present in the parasite nucleus for MPI and 6PGD, in contrast to the thousands of copies of the kinetoplast region that is amplified in kDNA-PCR [71] which confers higher sensitivity to the molecular detection of the parasite, regardless of the type of sample.

Unidentified species were associated with low parasite load independent of the sample type, similar to previous reports [17,33]. The results underline the utility of less invasive samples (L and FP versus Bx) for PCR diagnosis which is supported by a report that demonstrates that parasite load in superficial tissue is higher than inferior dermis and hypodermis [72].

The start of antileishmanial treatment does not necessarily depend on the knowledge of the infecting *Leishmania* species. However, it has been observed that certain species can influence treatment outcome [73,74], its failure [75] and the choice of treatment [9]. Furthermore, species identification has been highly recommended as part of clinical trials [76] and study of cutaneous leishmaniasis surveillance and outbreaks [77].

Similarly, the precision and reproducibility of the different sampling methods that guarantee a good performance of our molecular tests, has been key to the identification of *Leishmania (Viannia)* species in groups of sandflies described as putative vectors of the disease in the department of Madre de Dios, Peru [78–80]. Likewise, our study of population genetics and distribution of *L. (V.) braziliensis* in Peru was possible after the identification of the parasites [81]. Other studies where we evaluated two different types of rapid tests took advantage from the identification of *Leishmania* species for inclusion criteria such as one study that tested RPA-LF [23,82] and another on the CL-Detect device [83] both to be applied under field conditions. It should be noted that an additional advantage of our procedure is that the samples can be included in absolute ethanol at room temperature and thus be sent without the need for cold chain from the sampling sites to the laboratory where the molecular assays are carried out. Our findings support the use of minimal invasive sampling methods, such as scraping with lancets or filter paper impressions as alternatives to guarantee the correct identification of the parasite through molecular tests and thus ensure a more standardized management of the parasites mainly under field conditions.

## Conclusions

This study shows that the clinical sample collection procedure can directly influence the sensitivity of the FRET probe-based Nested Real-Time PCR that we developed to identify *Leishmania (Viannia)* species. In this sense, samples collected by scraping with lancets could be just as appropriate to identify the parasite in a similar way to filter paper and as an alternative to invasive methods such as biopsy. Standardized use of minimally invasive sampling methods is preferred for the benefit of the patient and potentially applicable to other molecular identification assays under field conditions.

## Supporting information

**S1 Fig. Flowchart that summarize all experimental procedures of this study.**
(TIF)

**S1 Table. Clinical data of study population.** All data were anonymized by coding and processed blindly as mentioned above. Legend for columns B to I: 1 positive result, 0 negative result, * weak/very weak amplification band in kDNA-PCR electrophoresis. Legend for column S, work: FFAA, Peruvian armed forces. Legend for column U, *Leishmania (Viannia)* species: BRA, *L. (V.) braziliensis*; GUY, *L. (V.) guyanensis*; LAIN, *L. (V.) lainsoni*; PER, *L. (V.) peruviana*; spp, *Leishmania* spp.; PER-BRA, possible hybrid *L. (V.) peruviana-L. (V.) braziliensis*; Neg, negative. The raw data in this table permit replicate the results of our study.
(XLSX)

## Acknowledgments

The authors would like to thank Edson Maguiña, Luis Angel Rosales and Dolores Rimarachín for their support in sample collection. Also, we want to thank Carmen Lucas for her invaluable administrative assistance. Finally, the authors are grateful to all patients who participated voluntarily in this study.

## Disclaimer

Some authors of this manuscript are employees of the U.S. Government. This work was prepared as part of their official duties. Title 17 U.S.C. §105 provides that "Copyright protection under this Title is not available for any work of the United States Government". Title 17 U.S.C. §101 defines a U.S. Government work as a work prepared by a military service member or employee of the U.S. Government as part of that person's official duties.

The views expressed in this article are those of the author and do not necessarily reflect the official policy or position of the Department of the Navy, Department of Defense, nor the U.S. Government.

## Author Contributions

**Conceptualization:** Maxy B. De los Santos, Erika S. Perez-Velez, Rocio del Pilar Santos, Ivonne Melissa Ramírez.

**Data curation:** Maxy B. De los Santos, Steev Loyola, Erika S. Perez-Velez, Rocio del Pilar Santos, Ivonne Melissa Ramírez, Hugo O. Valdivia.

**Formal analysis:** Maxy B. De los Santos, Steev Loyola, Erika S. Perez-Velez, Rocio del Pilar Santos, Ivonne Melissa Ramírez.

**Funding acquisition:** Hugo O. Valdivia.

**Investigation:** Maxy B. De los Santos, Steev Loyola, Rocio del Pilar Santos, Ivonne Melissa Ramírez.

**Methodology:** Maxy B. De los Santos, Steev Loyola, Erika S. Perez-Velez, Rocio del Pilar Santos, Ivonne Melissa Ramírez.

**Project administration:** Hugo O. Valdivia.

**Resources:** Maxy B. De los Santos.

**Software:** Steev Loyola, Ivonne Melissa Ramírez.

**Supervision:** Maxy B. De los Santos, Erika S. Perez-Velez, Rocio del Pilar Santos, Hugo O. Valdivia.

**Validation:** Maxy B. De los Santos, Steev Loyola, Erika S. Perez-Velez, Rocio del Pilar Santos, Ivonne Melissa Ramírez.

**Visualization:** Maxy B. De los Santos, Steev Loyola, Erika S. Perez-Velez, Rocio del Pilar Santos, Hugo O. Valdivia.

**Writing – original draft:** Maxy B. De los Santos, Rocio del Pilar Santos.

**Writing – review & editing:** Maxy B. De los Santos, Steev Loyola, Erika S. Perez-Velez, Rocio del Pilar Santos, Ivonne Melissa Ramírez, Hugo O. Valdivia.

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
