## [Decision Letter · Decision Letter 0]

18 Jan 2024

Dear PhD De los Santos,

Thank you very much for submitting your manuscript "Sampling is decisive to determination of Leishmania species" for consideration at PLOS Neglected Tropical Diseases. As with all papers reviewed by the journal, your manuscript was reviewed by members of the editorial board and by several independent reviewers. In light of the reviews (below this email), we would like to invite the resubmission of a significantly-revised version that takes into account the reviewers' comments. 

The Reviewers the Editors acknowledge the relevance of your report to the diagnosis and speciation of Leishmania and the importance/impact of the sampling methods used. The novelty of your report may be limited as a number of earlier studies have compared the various sampling methods for leishmania diagnosis, your study and the major findings are well documented and useful to the scientific community, especially to scientists interested in field diagnosis of leishmaniasis. Further, this report is well within the scope of PNTD. However, please address all the comments and questions of the reviewers in your revised manuscript. Specifically, please address in detail the comment related to false negativity mentioned by reviewer 1.

We cannot make any decision about publication until we have seen the revised manuscript and your response to the reviewers' comments. Your revised manuscript is also likely to be sent to reviewers for further evaluation.

Sincerely,

Alain Debrabant

Academic Editor

Walderez Dutra

Section Editor

The Reviewers the Academic Editor acknowledge the relevance of your report to the diagnosis and speciation of Leishmania and the importance/impact of the sampling methods used. The novelty of your report may be limited as a number of earlier studies have compared the various sampling methods for leishmania diagnosis, your study and the major findings are well documented and useful to the scientific community, especially to scientists interested in field diagnosis of leishmaniasis. Further, this report is well within the scope of PNTD. However, please address all the comments and questions of the reviewers in your revised manuscript. Specifically, please address in detail the comment related to false negativity mentioned by reviewer 1.

Reviewer's Responses to Questions

**Key Review Criteria Required for Acceptance?**

**Methods**

-Are the objectives of the study clearly articulated with a clear testable hypothesis stated?

-Is the study design appropriate to address the stated objectives?

-Is the population clearly described and appropriate for the hypothesis being tested?

-Is the sample size sufficient to ensure adequate power to address the hypothesis being tested?

-Were correct statistical analysis used to support conclusions?

-Are there concerns about ethical or regulatory requirements being met?

Reviewer #1: There is no hypothesis clearly stated, however the aims of the study are clearly articulated.

The design, population studied, and sample size are clearly described and appropriate; the statistical methods seem appropriate and there are no ethical concerns as consent was signed by the patients.

It is unclear though whether the study was blinded at any stage. This should be indicated, and the limitations of a non-blind study discussed.

Reviewer #2: See general comments

**Results**

-Does the analysis presented match the analysis plan?

-Are the results clearly and completely presented?

-Are the figures (Tables, Images) of sufficient quality for clarity?

Reviewer #1: The analysis and results are well described and supported.

The graph in figure 3B is difficult to read. Please make sure to increase the font and submit a high-quality image.

I also suggest creating a graphical abstract figure or a figure summarizing the various methods used and the main results yielded.

Reviewer #2: (No Response)

**Conclusions**

-Are the conclusions supported by the data presented?

-Are the limitations of analysis clearly described?

-Do the authors discuss how these data can be helpful to advance our understanding of the topic under study?

-Is public health relevance addressed?

Reviewer #1: The conclusions are supported and some limitations are mentioned.

The public relevance of the study is addressed.

Reviewer #2: (No Response)

**Editorial and Data Presentation Modifications?**

Reviewer #1: Minor revisions:

1) It is unclear though whether the study was blinded at any stage. This should be indicated, and the limitations of a non-blind study discussed. (See methods part)

2) The graph in figure 3B is difficult to read. Please make sure to increase the font and submit a high-quality image.(See results part)

3) I also suggest creating a graphical abstract figure or a figure summarizing the various methods used and the main results yielded. (See results part)

4) What is the prevalence in the territory of the various species of Leishmania identified in this article? This needs to be discussed in the introduction or discussion sections.

5) The authors talk about false negatives in the discussion. Is this also addressed in the data? This concept should be made more clear.

6) The authors should discuss why it is so important to determine the exact species of Leishmania affecting a certain individual (because of treatment choice, epidemiological research...etc?)

Reviewer #2: (No Response)

**Summary and General Comments**

Reviewer #1: This is a very interesting manuscript investigating different methods of Leishmania detection in order to improve species identification in the field. The results found will have a noticeable public health impact in the territory and could be extrapolated to other endemic areas as well. Overall, there are no major issues and I recommend publication with minor revisions. See my comments above.

Reviewer #2: In the paper described by de los santos et al, entitled “sampling is decisive to determination of Leishmania species, they compared the sensitivity of molecular identification of Leishmania species from Peru, using 3 types of sampling: punch biopsy, filter paper and lancet scraping. They showed that different sampling methods can affect the discrimination capacity of molecular tests for the identification of Leishmania species. They demonstrated that lancet scraping is more efficient than filter paper imprints or biopsies to identify more leishmanisais cases. They suggested that scraping with lancets could be used to identify parasite as an alternative to invasive methods such biopsy. It is very interesting study that deals with a topical problem, diagnosis of leishmaniasis, in fact the identification of the parasite are necessary for the efficiency of treatment. However, the authors need to improve the manuscript. 

Major points

1. The authors have to add controls: negative control samples from healthy individuals. They have to check the integrity of extracted DNA by using PCR targeting host gene like beta globin. It is very important to check the true negativity. It is also important to compare the recovery of DNA from each sampling methods, especially in the sensitive area (Eyes, lips…).

2. It would be interesting to consider the amplification and sequencing of another molecular marker for the identification of Leishmania species especially for those that were not identified. 

3. In the section 2 of results (line 285-line 296), 85 patients were identified infected by Leishmania, but in the section 3 (line 302-line313) and table 2 all the percentages were calculated based on the total 105 patients. In addition they have to add p value to show significant difference between sampling methods (line 307- line 308). 

4. Table 2 the parasite load is not clear, in my opinion they have to illustrate the parasite load separately in another figure.

5. For the weak band intensity, it could be interesting to add another molecular method for the identification of parasite.

Minor point

1. In my opinion the title is general, the authors should think about another title reflecting the message of the study

PLOS authors have the option to publish the peer review history of their article (what does this mean?). If published, this will include your full peer review and any attached files.

Reviewer #1: No

Reviewer #2: No
---

## [Decision Letter · Decision Letter 1]

8 Mar 2024

Dear PhD De los Santos,

Thank you very much for submitting your manuscript "Sampling is decisive to determination of Leishmania species" for consideration at PLOS Neglected Tropical Diseases. As with all papers reviewed by the journal, your manuscript was reviewed by members of the editorial board and by several independent reviewers. The reviewers appreciated the attention to an important topic. Based on the reviews, we are likely to accept this manuscript for publication, providing that you modify the manuscript according to the review recommendations. 

Please address the remaining two comments of Reviewer 2 in your revised manuscript.

Sincerely,

Alain Debrabant

Academic Editor

Walderez Dutra

Section Editor

Please address the remaining two comments of Reviewer 2 in your revised manuscript.

Reviewer's Responses to Questions

**Key Review Criteria Required for Acceptance?**

**Methods**

-Are the objectives of the study clearly articulated with a clear testable hypothesis stated?

-Is the study design appropriate to address the stated objectives?

-Is the population clearly described and appropriate for the hypothesis being tested?

-Is the sample size sufficient to ensure adequate power to address the hypothesis being tested?

-Were correct statistical analysis used to support conclusions?

-Are there concerns about ethical or regulatory requirements being met?

Reviewer #1: Yes. The authors have responded to my comments and addressed my concerns.

Reviewer #2: (No Response)

**Results**

-Does the analysis presented match the analysis plan?

-Are the results clearly and completely presented?

-Are the figures (Tables, Images) of sufficient quality for clarity?

Reviewer #1: The results are clearly presented, more controls were added, and the figure quality has increased.

Reviewer #2: (No Response)

**Conclusions**

-Are the conclusions supported by the data presented?

-Are the limitations of analysis clearly described?

-Do the authors discuss how these data can be helpful to advance our understanding of the topic under study?

-Is public health relevance addressed?

Reviewer #1: The conclusion section is satisfactory.

Reviewer #2: (No Response)

**Editorial and Data Presentation Modifications?**

Reviewer #1: /

Reviewer #2: (No Response)

**Summary and General Comments**

Reviewer #1: /

Reviewer #2: The authors answered to all queries and included all the modifications requested. They improved the quality of the manuscript. Two points still need to be revised. 

Minor revision

1. The title is still general, not limited to what was carried out. Indeed the authors used just Leishmania viannia species causing CL in Peru. I suggest ex “sampling is decisive to Leishmania viannia species identification…

2. The authors answered that they used samples from healthy individuals as a negative controls. Please mention that in the text in MM (study population).

PLOS authors have the option to publish the peer review history of their article (what does this mean?). If published, this will include your full peer review and any attached files.

Reviewer #1: No

Reviewer #2: No

Figure Files:

Data Requirements:

Reproducibility:

References

---

## [Editor Report · Decision Letter 2]

28 Mar 2024

Dear PhD De los Santos,

We are pleased to inform you that your manuscript 'Sampling is decisive to determination of Leishmania (Viannia) species' has been provisionally accepted for publication in PLOS Neglected Tropical Diseases.

Best regards,

Alain Debrabant

Academic Editor

Igor C. Almeida

Section Editor

---

## [Editor Report · Acceptance letter]

7 Apr 2024

Dear PhD De los Santos,

We are delighted to inform you that your manuscript, "Sampling is decisive to determination of * Leishmania (Viannia)* species," has been formally accepted for publication in PLOS Neglected Tropical Diseases.

Best regards,

Shaden Kamhawi

co-Editor-in-Chief

Paul Brindley

co-Editor-in-Chief
